# Collective geographical eco-regions and precursor sources driving Arctic new particle formation

James Brean[1], David C.S. Beddows[1], Roy. M. Harrison[1&], Congbo Song[1], Peter Tunved[2], Johan Ström[2], Radovan Krejci[2], Eyal Freud[2], Andreas Massling[3], Henrik Skov[3], Eija Asmi[4], Angelo Lupi[5] and Manuel Dall'Osto[6]

[1]Division of Environmental Health & Risk Management School of Geography, Earth & Environmental Sciences University of Birmingham, Edgbaston, Birmingham, B15 2TT, United Kingdom
[2]Department of Environmental Science & Bolin Centre of Climate Research, Stockholm University, Stockholm 10691, Sweden
[3]Interdisciplinary Centre for Climate Change (iClimate), Department of Environmental Science, Aarhus University, Roskilde 4000, Denmark
[4]Atmospheric Composition Research, Finnish Meteorological Institute, Helsinki, Finland
[5]Institute of Polar Science, CNR, Bologna, Italy
[6]Institute of Marine Science, Consejo Superior de Investigaciones Científicas (CSIC), Barcelona, Spain
[&]Also at: Department of Environmental Sciences / Center of Excellence in Environmental Studies, King Abdulaziz University, PO Box 80203, Jeddah, 21589, Saudi Arabia

*Correspondence to*: M.D.O: dallosto@icm.csic.es; J.B: j.brean@bham.ac.uk

**Abstract.** The Arctic is a rapidly changing ecosystem, with complex ice-ocean-atmosphere feedbacks. An important process is new particle formation (NPF) from gas phase precursors, which provide a climate forcing effect. NPF has been studied comprehensively at different sites in the Arctic ranging from those in the high Arctic, those at Svalbard, and those in the continental Arctic, but no harmonized analysis has been performed on all sites simultaneously, with no calculations of key NPF parameters available for some sites. Here, we analyse the formation and growth of new particles from six long-term ground-based stations in the Arctic (Alert, Villum, Tiksi, Mt. Zeppelin, Gruvebadet. & Utqiagvik). Our analysis of particle formation and growth rates, as well as back trajectory analysis shows summertime maxima in frequency of NPF and particle formation rate at all sites, although the mean frequency and particle formation rates themselves vary greatly between sites, highest at Svalbard, and lowest in the high Arctic. Summertime growth rate, condensational sinks and vapour source rates show a slight bias towards the southernmost sites, with vapour source rates varying by around an order of magnitude between the northernmost and southernmost sites. Air mass back trajectories during NPF at these northernmost sites are associated with large areas of sea ice and snow, whereas events at Svalbard are associated with more sea ice and ocean regions. Events at the southernmost sites are associated with large areas of land, and sea ice. These results emphasize how understanding the geographical variation in surface type across the Arctic is key to understanding secondary aerosol sources and provide a harmonised analysis of NPF across the Arctic.

# 1. Introduction

Earth's changing climate is substantially increasing temperatures in the Arctic (IPCC, 2014), resulting in loss of sea ice and unprecedented melting of the Greenland ice sheet. Atmospheric aerosols are known to impact the Arctic radiation balance directly (e.g., Sand et al., 2017, Najafi et al., 2016) and alter Arctic clouds (Garrett et al., 2002; Garrett and Zhao, 2006). The impact of aerosol over the last century has been to cool the Arctic (Shindell and Faluvegi 2009, Acosta Navarro et al. 2016). Thus, there is an urgent need to accurately model this Arctic aerosol to constrain climate sensitivity estimates and predict future patterns in aerosol distribution and sources within the Arctic. However, atmospheric chemical transport and climate models consistently fail to replicate much of the observed variation of aerosol concentrations observed at ground-based stations (Sand et al., 2017) and recently there has been shown a different temporal trend in predicted and observed cloud cover at a high Arctic site (Gryning et al. 2021) continuing a historical overestimation of low-level clouds in the Arctic by climate models, particularly in the wintertime, most particularly in daily data (Quan et al., 2012). Uncertainties also exist in cloud coverage retrieved by satellites, arising from similarities between clouds and ice-snow surfaces, and frequent temperature and humidity inversions (Browse et al., 2012)

Different measurements at Arctic sites show a strong annual cycle in aerosol characteristics, largely dictated by new particle formation (NPF) (Tunved et al., 2013; Dall'Osto et al., 2017a; 2018a; 2018b), a process characterised by a sudden burst of nanometre sizes particles in the atmosphere, followed by their growth to larger sizes. The initial formation of these particles is driven by the clustering of gases in the atmosphere to form clusters at a rate faster than their losses due to evaporation or condensation, the second step is driven by both coagulation, and condensation of vapours with sufficiently low vapour pressures to condense down on new particles (Lee et al., 2019). NPF is estimated to be responsible for around half of global cloud condensation nuclei (CCN) concentrations when neutral and ion-induced mechanisms of particle formation from sulphuric acid, ammonia, and organics are considered (Gordon et al., 2017). These models neglect mechanisms such as iodine nucleation shown to be important in the high Arctic (Baccarini et al., 2020), amines as stabilisers, also important in Antarctica (Brean et al., 2021), and particle formation involving nitric acid, important in the upper troposphere (Wang et al., 2022). Other models neglect all mechanisms except those involving sulphuric acid (Liu et al., 2012). As particle concentrations in the Arctic are generally very low, cloud properties in the region are sensitive to small perturbations to CCN counts (Birch et al., 2012). Early measurements of particle size distributions in the Arctic pointed towards an important and highly variable source of nucleation mode aerosol (Covert et al., 1996), with indication that these particles were produced above or in upper layers of the marine boundary layer, or from precursors emitted by the open sea (Wiedensohler et al., 1996). Prior research points towards NPF as a summertime phenomenon (Asmi et al., 2016; Croft et al., 2016; Freud et al., 2017; Leaitch et al., 2013; Nguyen et al., 2016; Tunved et al., 2013). Recently, two papers using on-line mass spectrometric instrumentation to probe the first steps of cluster formation in Arctic NPF have been published, showing NPF being driven by iodine oxoacids at Villum research station & the central Arctic ocean (Baccarini et al., 2020; Beck et al., 2020), while NPF at Svalbard was driven by the

oxidation products of dimethylsulphide (DMS), alongside ammonia with a contribution of oxygenated organic molecules in the summertime (Beck et al., 2020). Moschos et al. (2022) recently performed a pan-Arctic analysis of organic aerosol (OA), highlighting an annual cycle where decreasing anthropogenic emissions in the summer are replaced by natural aerosol sources, leading to a relative uniformity in annual OA concentrations. These natural aerosol sources are largely secondary, and they show that the biogenic secondary organic aerosol concentrations are highly sensitive to temperature changes.

NPF events are dependent upon the precursor vapour concentrations, temperature, ion pair production rate, and the surface area of pre-existing aerosols, and thus this CCN contribution varies regionally, and is a result of an interplay between these factors (Lee et al., 2019). The key parameters driving NPF in the Arctic are not well understood. In polluted locations the surface area of pre-existing particles often dictates NPF occurrence (Lee et al., 2019). However, in remote locations condensation sinks are consistently low (Sellegri et al., 2019), and concentrations of precursors and solar radiation intensity may be key in dictating NPF frequency and intensity. However, with multiple potential mechanisms and many poorly understood sources of precursors from the many and varied eco-regions, Arctic NPF demands further study. Broadly, here we define an eco-region as an ecologically and geographically defined area that captures not only the distribution of biological communities but also the environmental conditions (including climate variables) such as ice sheet, marginal sea-ice zone, tundra, snow-covered land, sea-ice influenced open ocean, permanent open ocean, animal-colonised shores and islands, etc. (Barry et al., 2013; Meltorft et al., 2013; CAFÉ 2017; Schmale et al., 2021). Research at different sites has, for example, pointed towards sea ice (Allan et al, 2015; Baccarini et al, 2020; Dall'osto et al., 2017b; Dall'osto et al., 2018b; Heintzenberg et al., 2015), and open water (Dall'osto et al., 2018b; Croft et al., 2019; Wiedensohler et al., 1996; Willis et al., 2017) regions as sources of new particle precursors.

Linking eco-regions and new particle formation highlights that an emphasis on source processes, and their interplay with atmospheric physical and photochemical conditions is crucial to understand the driving forces behind Arctic NPF. Despite the numerous long-term measurements which have been conducted for many years, comparisons of NPF events between many Arctic sites remain sparse (Freud et al., 2017; Dall'Osto et al., 2019b). Motivated by the lack of studies comparing NPF at these sites simultaneously, and following in the stead of previous publications studying multiple sites in different environments (e.g., Sellegri et al., 2019) with calculations of the key parameters of particle formation and growth rates, we used long term coordinated field measurement studies of aerosol size distribution to manually identify NPF events by the time evolution of the particle size distribution across the Arctic, investigating the rates of particle formation and growth. We further used back-trajectory analysis to determine the air masses associated with NPF events, how strongly each trajectory contributed to NPF, and the surface types these air masses flowed over prior to NPF occurrence (open ocean, sea ice, land, or snow). Our results show that bursts of newly formed particles in different Arctic regions are associated with different source regions, indicating the likelihood of multiple mechanisms at play.

## 2. Methods

### 2.2. Sampling sites

Aerosol particle size distributions were collected from six Arctic long-term sites summarized in Table 1, with data coverage is shown in Fig. 1 after being filtered for anthropogenic influences based on either the shape of the size distribution, or air mass direction (Asmi et al., 2016; Freud et al., 2017, Dall'Osto et al., 2018a,b). There is limited data overlap between the sites, with best overlap during 2015, where data is measured for several months at all sites except one. The mean size distribution from each site for this period, alongside the mean across all time periods is plotted in Fig S1. The location of each site is highlighted in Fig. S2. Data from The Dr. Neil Trivett Global Atmosphere Watch Observatory at Alert (ALE) was collected using a TSI 3034 SMPS (Steffen et al., 2014), representing the Northernmost site, 8 km from the shore of Ellesmere Island. This site represents the Canadian Arctic Archipelago. The Villum Research Station is located near Station Nord (VRS) and is located in Northeast Greenland 3 km from the shore. A Vienna-type DMA attached to a TSI 3772 CPC collects the size distribution. These are the two northernmost sites and spend most of the years surrounded by sea ice. The measurement site at Tiksi (TIK) is in the Russian Arctic, 500 m from the shore and 5 km from the town of Tiksi itself. A twin DMPS system collects the size distribution here. The Utqiagvik measurement site (UTQ, formerly known as Barrow) is 3 km from the shore, and 5 km northeast of the nearest town in Alaska, where a custom-built DMPS connected to a TSI 3772 CPC collects the size distribution data. Data coverage here is relatively low (25%) due to regular anthropogenic influence. Together, these two sites represent the continental Arctic. The measurement site at Gruvebadet (GRU) is in proximity of the village of Ny-Ålesund, southeast of the main agglomerate of the village. A TSI 3034 SMPS measures the size distribution here. Measurements at Mt. Zeppelin (ZEP) are conducted at a height of 474 m, adjacent to the GRU measurement site. A size distribution is collected here using a custom twin DMPS system with one Vienna-type medium DMA coupled to a TSI 3010 CPC, and a Vienna-type short DMA coupled to a TSI 3772 CPC. The aerosol dynamics at both of these sites are quite dissimilar due to the ~400 m elevation difference and differing prevailing winds (Dall'Osto et al., 2019b). Intercomparison workshops have shown differences between instruments measuring particle size distributions to be within 10%, increasing at smaller diameters (Wiedensohler et al., 2012). This produces some uncertainty when we are comparing particle formation rates and growth rates of particles in these smaller size regimes, but this uncertainty is substantially smaller than the differences in particle concentrations between sites.

Throughout this text, the seasons are defined as spring (MAM), summer (JJA), autumn (SON) and winter (DJF). All times are in local time. All data were cleaned and filtered as described in Asmi et al., 2016; Freud et al (2017), and Dall'Osto et al., (2017, 2018a, b). Overall, our large dataset is composed of 9765 days of SMPS size distributions collected at ALE (972 days), UTQ (594 days), GRU (1019 days), ZEP (3356 days), VRS (1735 days) and TIK (1999 days). The size distribution from 10 nm is used for all sites. At the TIK site there is occasionally missing data in the 10 - 16 nm range. Formation and growth rates derived from these data are not used here.

**Table 1: List of sampling sites and NPF events (689 in total). We separate sites into "types" based upon the classification given by Schmale et al., (2021). Two ranges for the size range are given for TIK as the second CPC was changed from the TSI 3772 to the 3776.**

| Site | Lat | Lon | Elevation a.s.l (m) | Type | Number of NPF events | Instrumentation, size range (nm) |
|------|-----|-----|---------------------|------|----------------------|----------------------------------|
| Alert (ALE) | 82.5 | -62.3 | 210 | Archipelago | 20 | TSI 3034 SMPS, 10 – 470 |
| Nord (VRS) | 81.6 | -16.7 | 24 | Greenland | 64 | Custom DMPS, TSI 3772 DMA, 9 - 916 |
| Gruvebadet (GRU) | 78.9 | 11.9 | 67 | Svalbard | 155 | TSI 3034 SMPS, 10 – 470 |
| Zeppelin (ZEP) | 78.9 | 11.9 | 474 | Svalbard | 251 | Custom Twin-DMPS, TSI 3010, TSI 3772, 10 – 750 |
| Utqiagvik (UTQ) | 71.3 | -156.6 | 5 | Continental | 31 | Custom DMPS, TSI 3772, 9 - 985 |
| Tiksi (TIK) | 71.6 | 128.9 | 35 | Continental | 168 | Custom Twin-DMPS, TSI 3772, TSI 3776, 10-800, 3-100 |

## 2.2 NPF parameters

The condensation sink (CS, $s^{-1}$) represents the rate at which a vapour phase molecule will collide with pre-existing particle surface, and was calculated from the size distribution data as follows (Kulmala et al., 2012):

$$CS = 2\pi D \sum_{d_p} \beta_{m,d_p} d_p N_{d_p} \ (1)$$

where D is the diffusion coefficient of the diffusing vapour (assumed sulphuric acid), $\beta_m$ is a transition regime correction (Kulmala et al., 2012), $d_p$ is particle diameter, and $N_{d_p}$ is the number of particles at diameter $d_p$. The formation rate of new particles at size dp (Jdp) is calculated as follows, presuming a homogeneous airmass:

$$J_{d_p} = \frac{dN_{d_p}}{dt} + CoagS_{d_p} . N_{d_p} + \frac{GR}{\Delta d_p} . N_{d_p} \ (2)$$

where the first term on the right-hand side comprises the rate at which particles enter the size $d_p$, and the second term refers to losses from this size by coagulation (CoagS$_{dp}$ being the coagulation sink at size $d_p$, and $N_{dp}$ being the number of particles at size $d_p$, calculated according to Cai & Jiang, 2017), with the third term referring to losses from this size by growth, where the growth rate of new particles is as follows:

$$GR = \frac{ddp}{dt} \ (3)$$

This was calculated through the lognormal distribution method outlined in Kulmala et al., (2012). If it is presumed that all particle growth is driven by sulphuric acid condensation, then the condensing vapour concentration needed to describe the observed particle growth rates ($C_{vap}$) can be calculated thus (Nieminen et al., 2010):

$$C_{vap} = \frac{\rho_p}{D_v m_v \Delta t}\left(\frac{d_p{}^2 - d_{p0}{}^2}{8} + \left(\frac{4}{3\alpha} - 0.623\right)\frac{\lambda}{2}(d_p - d_{p0}) + 0.623\lambda^2 ln\frac{2\lambda + d_p}{2\lambda + d_{p0}}\right) \ (4)$$

Where $\rho_p$ is the particle phase density, $D_v$ is the diffusivity of vapour (sulphuric acid), $m_v$ is the mass of one molecule of vapour, $d_p$ and $d_{p0}$ are the particle diameters at times $t$ and $0$ respectively, $\lambda$ is the mean free path of the condensing vapour (sulphuric acid) and $\alpha$ is the mass accommodation coefficient (presumed to be 1). The steady-state production rate (Q) of this vapour is therefore described by the product of the concentration and loss terms (Dal Maso et al., 2005):

$$Q = C_{vap} \cdot CS \ (5)$$

In equations (4) and (5), the assumption is of course that sulphuric acid is the sole vapour driving particle growth. Across the Arctic, MSA, sulphuric acid, ammonia, and iodine oxides have all been shown to contribute to particle growth (Beck et al., 2020), however, as the condensed phase density and molecular masses of these molecules vary widely, thus for these calculations we make the assumption that sulphuric acid drives all growth, but note that this is a source of uncertainty.

## 2.3 Analysis of NPF events

NPF events, identified visually based on the time evolution of the time evolution, here plotted as contour plots using the criteria of Dal Maso et al. (2005) were separated into 3 types by manual inspection: type A represents events with formation and particle growth ("Banana" type events), type B represents events with limited growth (particles do not make it to 30 nm) and type C represents events where the particles appear at >10 nm, here presumed to be particles advected from a nearby location, where new particles have formed and the new mode of particles is growing at the time of measurement. Only types A and B are used in our data analysis. Formation of particles at the smallest measured sizes is a key characteristic of NPF and is required to calculate formation rates reliably. There is also a chance that Type C events include particles not formed secondarily, but just shows growth of primary particles, and thus we neglect to include Type C events in these analyses. These events were isolated and classified by the shape of their size distributions. Examples are shown in Fig. S3, NPF event start and stop points are shown in Fig. S4.

## 2.4 Back trajectories & concentration weighted trajectories

The NOAA HYSPLIT model was used to calculate 3 day back-trajectories for air masses arriving at the sampling sites. Each back-trajectory data point was assigned to a surface-type (land, sea, ice, or snow over land. A cell is considered ice-covered if more than 40 % of the cell is covered with ice) on a 24 km grid from the daily Interactive Multisensor Snow and Ice Mapping System (IMS) (Anon, US National Ice Center, 2008). To investigate sources leading to particle growth, these 72-hour back-

trajectories were gridded to 1x1 grid cells of 1 degree each, and linked back to the steady-state production rate of equivalent sulphuric acid by the following equation:

$$ln(\bar{C}_{ij}) = \frac{1}{\sum_{k=1}^{N} \tau_{ijk}} \sum_{k=1}^{N} ln(c_k)\tau_{ijk} \quad (6)$$

Where $\bar{C}_{ij}$ is the concentrated weighted trajectory at cell $i, j$, $N$ is the total number of trajectories, $c_k$ is the value of $Q$ associated with the arrival of trajectory $k$, and $\tau_{ijk}$ is the residence time of trajectory $k$ in grid cell $i, j$. $\bar{C}_{ij}$ therefore describes the source strength of condensable vapour that drives particle growth from any particular grid cell (Hsu et al., 2003; Lupu and Maenhaut, 2002). This was done using the trajLevel function in the Openair package in R 3.4.3. Trajectories more than 1,000 m a.g.l. were not considered in these analyses, excluding 2.2% of trajectories, mostly at ZEP . 72 hours lies somewhere between the long atmospheric lifetime of $SO_2$ (von Glasow et al., 2009), and the shorter lifetime of reactive VOCs, MSA, and iodine compounds (Fuentes et al., 2000; Sherwen et al., 2016; Kloster et al., 2006).

# 3 Results

## 3.1 Seasonal variation of NPF

Fig. 2 shows the characteristics of NPF events by month for all sites, each site weighted by the span of data available. $J_{10}$ values peak in the summertime, with summertime means significantly greater than the mean for other seasons (0.14 $cm^{-3}$ $s^{-1}$ in summer, 0.054 $cm^{-3}$ $s^{-1}$ through other seasons, frequency >10% for the months JJA), coincidental with the months of highest insolation, and likely those of highest photochemical activity. Growth rates are also higher in summer months compared to the mean for other seasons (1.6 nm $h^{-1}$ in summer, 0.93 nm $h^{-1}$ through other seasons). While winter and springtime periods are typically associated with higher accumulation mode loading due to Arctic Haze (Abbatt et al., 2019; Asmi et al., 2016; Heintzenberg et al., 2017), no significant difference is seen in CS between seasons during NPF periods, although individual months vary by over a factor of 3 (Fig. 2). CS between NPF and non-NPF events across the whole year is also similar (mean CS 8.6·$10^{-4}$ $s^{-1}$ and 8.9·$10^{-4}$ $s^{-1}$ during NPF and non-NPF periods, respectively). It is worth note that wintertime NPF, although making up a small number of total NPF events, tends to occur under a 15% lower mean CS than the seasonal average.

In contrast to particle formation rates, source vapour rates do not have a clear seasonal trend, but when averaged across seasons, source vapour rates do show lower source rates in winter (3.2·$10^4$ $cm^{-3}$ $s^{-1}$ compared to 1.6·$10^4$ $cm^{-3}$ $s^{-1}$ through other seasons). Wintertime events were observed at all sites except the two most northerly ones (ALE and VRS). The southernmost sites experience more wintertime insolation, possibly explaining the lack of NPF at these northern sites. Events most frequently started between 9:00 & 12:00, with the visual signature of an ongoing NPF event visible in the measured size distributions for slightly under 12 hours (median, Fig. S4), despite the fact that these sites are often in 24 hour sunlight.

As $J_{10}$ measures the rate of particles forming at 10 nm, it is also highly sensitive to the rates of coagulation between 1.5 – 10 nm. Coagulation rates in this range are much greater than coagulation rates at larger sizes and must be outcompeted by particle growth rates for an NPF event to be visible in the datasets analysed here. Events where particles fail to reach 10 nm will not be measured by the particle counting systems employed here. Such events have been reported during iodine-driven NPF events at similar latitudes (Baccarini et al., 2020; Beck et al., 2021); thus NPF frequency at these sites may well be higher than is reported here.

## 3.2 Spatial variation of summertime NPF features

The site-by-site variation in summertime NPF event characteristics is shown in Fig. 3. The CWTs weighted by Q for each site are plotted in Fig. 4 (CWTs across the whole Arctic region in Fig. S5), indicating source regions of equivalent sulphuric acid vapour leading to particle growth. Equation (5) essentially gives the interplay between the concentration of equivalent sulphuric acid driving particle growth ($C_{vap}$, calculated from the particle growth rate) and the loss of this vapour (CS). The land surface types (land, sea, ice, or snow over land) which 72-hour back trajectory points arriving at the site during NPF events flow over are plotted in Fig. 5, showing the surface-types that air masses flow over which led to NPF events. Here, ALE and VRS are discussed together as "high Arctic", as both of these sites are high latitude sites with similarly low $J_{10}$, GR, and CS values (Fig. 3), GRU and ZEP are talked about together as "Svalbard" sites as they are co-located and surrounded by the same open and ice containing ocean, with similar $J_{10}$, GR, and CS, and although dissimilar in $J_{10}$, GR, and CS, the low latitude TIK and UTQ are seen to represent the "continental Arctic". Figure S1 shows the average size distribution during the period March – July 2015 where data was being collected at all sites except ALE, where the data for March – July 2013 is shown. All sites have two distinct modes, an Aitken mode peaking somewhere between 20 to 50 nm, and an accumulation mode peaking somewhere from 100 to 200 nm. The average across all periods is also shown, for which the distributions are similar, except ZEP which compared to the whole period of data availability, has a substantially larger mode at ~20 nm in this 2015 period. The size distribution at ALE and VRS shows overall low particle counts, especially at ALE. The two Svalbard sites, GRU and ZEP have similar size distributions, while those at TIK show a large Aitken mode, and UTQ shows a large accumulation mode.

### 3.2.1 High Arctic sites

Fig. 3 shows that NPF occurs at lower $J_{10}$ at the high latitude site ALE compared to the 4 lower latitude sites, but similarly at VRS compared to these other sites ($1.9 \cdot 10^{-2}$ & $5.0 \cdot 10^{-2}$ cm$^{-3}$ s$^{-1}$ at ALE and VRS, respectively, average for other sites $4.9 \cdot 10^{-2}$ cm$^{-3}$ s$^{-1}$). These two sites show lower GRs (0.69 & 0.73 nm h$^{-1}$ at ALE and VRS, respectively, average for other sites 1.35 nm h$^{-1}$) and at lower CS than the other sites ($1.8 \cdot 10^{-4}$ s$^{-1}$ & $3.0 \cdot 10^{-4}$ s$^{-1}$ at ALE and VRS, respectively, average for other sites $7.5 \cdot 10^{-4}$ s$^{-1}$), resulting in substantially lower Q values ($2.3 \cdot 10^{3}$ cm$^{-3}$ s$^{-1}$ & $4.2 \cdot 10^{3}$ cm$^{-3}$ s$^{-1}$ at ALE and VRS, respectively, average for other sites $2.6 \cdot 10^{4}$ cm$^{-3}$ s$^{-1}$). NPF at ALE occurs on 5.1% of days, particle formation at VRS occurs on 8.6 % of days (mean

NPF frequency across all other sites 17.1%). Of all Arctic sites, NPF is most infrequent at these high Arctic sites. Previous reports of Arctic iodine-NPF events report similarly low growth rates (Baccarini et al., 2020; Beck et al., 2021). Particle formation at GRU has been shown to be driven by clustering of $H_2SO_4$, methane sulfonic acid (MSA) and $NH_3$, with rapid particle growth driven by HOMs in the summertime (Beck et al., 2021).

The CWTs show air masses associated with high Q values at ALE arise from the surrounding Canadian Arctic Archipelago, and the western coast of Greenland, and those for particle growth at VRS have a strong source from the western coast of Greenland also, with some sources from mainland Greenland, and the surrounding iced and non-iced oceans (Fig. 4, 5, S5). Back trajectory analyses show NPF at both of these sites occur under air masses flowing over regions of snow and sea ice (71.4% & 80.4% of NPF 72 hour back trajectory datapoints flowing over sea and ice combined for ALE and VRS respectively). Notably, air masses arriving at ALE during NPF events are associated with markedly more sea ice surface than in non-NPF events (29 % during non-NPF periods versus 50 % during NPF periods). Air mass surface types for VRS do not change much between NPF and non-NPF periods.

### 3.2.2 Svalbard sites

At the two sites located at Svalbard, particle GRs were similar to one another, (0.92 & 1.0 nm $h^{-1}$ at GRU and ZEP, respectively, other sites ranging from mean GRs of 0.8 – 2.6 nm $h^{-1}$, with average 1.2 nm $h^{-1}$). $J_{10}$ is higher at the Svalbard sites than other sites (6.9·$10^{-2}$ & 6.1·$10^{-2}$ cm$^{-3}$ s$^{-1}$ at GRU and ZEP, respectively, average for other sites 3.4·$10^{-2}$ cm$^{-3}$ s$^{-1}$). Q here is greater than the high Arctic sites, but lower than the lower latitude continental Arctic sites (1.5·$10^4$ & 1.6·$10^4$ cm$^{-3}$ s$^{-1}$ at GRU and ZEP, respectively, average for other sites 2.0·$10^4$ cm$^{-3}$ s$^{-1}$). Similarly, CS is greater than the high Arctic sites, TIK, and similar to UTQ (7.1·$10^{-4}$ s$^{-1}$ & 6.3·$10^{-4}$ s$^{-1}$ at GRU and ZEP, respectively, average for other sites 5.4·$10^{-4}$ s$^{-1}$). The NPF frequency is lower at the higher altitude ZEP (18.3%) site compared to GRU (23.4%). The average NPF frequency at the other sites is 11.8%)

This Svalbard region is surrounded by open water due to advection of warm Atlantic water, and the CWTs for Q point to potential source regions of precursor vapours for particle growth from all the surrounding open ocean and sea ice regions (37.0% & 42.8% open ocean, and 43.5 & 40.3% sea ice regions for GRU and ZEP respectively, Fig. 4, 5, S5), indicating air masses driving NPF are not from one ocean region.

### 3.2.3 Continental sites

The sites TIK and UTQ represent the continental Arctic, being the southernmost sites (71.6° & 71.3°, located in Russia and Alaska respectively). $J_{10}$ at these southern sites do not differ greatly from the mean (3.9·$10^{-2}$ & 2.9·$10^{-2}$ cm$^{-3}$ s$^{-1}$ at TIK and UTQ, respectively, average for other sites 5.0·$10^{-2}$ cm$^{-3}$ s$^{-1}$). Particle GRs at these continental sites are the highest of all Arctic sites, especially at TIK (2.2 & 1.1 nm $h^{-1}$ at TIK and UTQ, respectively, average for other sites 0.84 nm $h^{-1}$). These high growth

rates make TIK a distinct site in the Arctic, CS during NPF events at TIK are greatest in the entire dataset. CS at UTQ is comparable to the Svalbard sites ($1.0 \cdot 10^{-3}$ s$^{-1}$ & $6.9 \cdot 10^{-4}$ s$^{-1}$ at TIK and UTQ, respectively, average for other sites $4.5 \cdot 10^{-4}$ s$^{-1}$). Q values at these sites (especially TIK) are, due to the high particle growth rates, high ($5.4 \cdot 10^{4}$ & $2.0 \cdot 10^{4}$ cm$^{-3}$ s$^{-1}$ at TIK and UTQ, respectively, average for other sites $9.4 \cdot 10^{3}$ cm$^{-3}$ s$^{-1}$)

The CWT and land type analysis indicates that the source region most strongly associated with high values of Q at TIK is the continental regions surrounding the sampling site, air masses during NPF spending 58.6% of time over land regions, elevated greatly from 18.9% during non-NPF event periods, indicating terrestrial sources of NPF precursors, rather than the marine. These events are unique compared to the open water, coastal and sea ice influenced NPF events observed at the other sites. This is shown in Fig. 5, showing that events are dominated by snow free land-based sources. Similarly, at UTQ the strongest vapour source is in the direction of the closest oil fields to the west. This region has been shown to be a driver of particle growth (Kolesar et al., 2017), although the back trajectory analysis shows that most of the air masses during NPF events are sea-ice dominated (80.1%).

## 4. Discussion

The Arctic is a highly geographically and biologically diverse region and understanding the drivers of NPF involves understanding a vast network of gas and aerosol sources and sinks. The results reported in this paper highlight the seasonal variation in Arctic NPF (Fig. 2), as well as the variation between different measurement sites during the summertime with $J_{10}$, GR, CS, and Q varying by orders of magnitude between sites (Fig. 3). The site-by-site variation in CS, J, and Q were tabulated recently in a review paper by Schmale and Baccarini (2021), to which our calculated values are similar where similar numbers are available (Figs. 2 and 3). For the sites where figures are not available, we provide the first reports of key NPF parameters. These results cover a multi-year period across the Arctic. We highlight that some of these sites have limited data coverage (Figure 1) and the periods of data overlap between sites are limited, although the size distributions for these periods of overlap are similar to the average across all periods (Figure S1). We also note the inherent uncertainty in particle size distribution measurements between sites, especially in both the <20 nm size range, which is particularly important to these NPF studies (Wiedensohler et al., 2012).

We show that the vapours which drive particle growth at each of these sites often (but not always) coincide with air masses flowing over particular, directional source regions (Fig. 4). NPF in the Arctic atmospheric boundary layer is occurring within air masses flowing over vastly different Arctic eco-regions, these being regions of open ocean water, consolidated and open pack ice, snow-covered land, and non-snow-covered land (Fig. 5), reflected in the variety of mechanisms to be seen in molecular scale measurements of new particle formation and growth (Baccarini et al., 2020; Beck et al., 2020). This variability

as it relates to NPF mechanisms has been highlighted in recent papers (Schmale and Baccarini, 2021). We highlight this complex network of NPF precursor sources in Fig. 6.

NPF at the northernmost sites (VRS, ALE) occurs when air masses arriving at the site have flown over regions of ice and snow. The slower rates of particle formation here are consistent with recent detailed reports of particle formation and growth in this region using on-line mass spectrometry (Baccarini et al., 2020; Beck et al., 2020), showing NPF driven by iodine oxoacids. Iodine has been shown to accumulate in algae (Küpper et al., 2008), which may be plentiful in the microalgal aggregates within the iced sympagic Arctic regions (Assmy et al., 2013; Boetius et al., 2013). Thinning of sea ice has already

caused an increase to atmospheric iodine levels (Cuevas et al., 2017). Future sea ice melt may accelerate NPF in this region due to enhanced precursor emissions. The CWT analysis here also shows a strong vapour source arising from the coast of Greenland. Source apportionment studies applied to highly time resolved VOC data show coastal Greenland to be a dominant source of DMS (Pernov et al., 2021), as well as ammonia from seabird colonies (Riddick et al., 2012). Arctic melt ponds, leads and melting ice are also sources of DMS (Levasseur 2013), thus a further influence of DMS oxidation products is feasible. If

iodine oxoacids are not the species responsible for particle formation observed in this dataset, these low formation rates may be related to low concentrations of alternative precursors, and weak solar radiation reducing both rates of photochemistry and ion pair production.

Events at the Svalbard sites (ZEP, GRU) occur within air masses flowing over regions of open ocean and iced ocean. NPF in

this region has been shown to be driven by sulphuric acid, ammonia, and oxygenated organic molecules (Beck et al., 2020). A main aerosol precursor from the open ocean is DMS. Emissions of DMS are increasing due to reductions in sea-ice extent (Galí et al., 2019), with DMS being an important source of both methanesulphonic acid ($CH_3SO_3H$, MSA) and sulphuric acid ($H_2SO_4$) (Hoffmann et al., 2016, Park et al., 2017; Kecorius et al., 2019, Park et al., 2021; Jang et al., 2021; Lee et al., 2020); further, open water regions are a source of oxygenated organic compounds (Mungall et al., 2017). Modelling studies

demonstrate a contribution of marine secondary organic aerosol to the total size distribution (Croft et al., 2019).

NPF at the Russian continental TIK site is heavily influenced by air masses flowing over land. Recent biogenic volatile organic compound emission data from Arctic tundra, sub-arctic wetland, underlain by discontinuous permafrost have been reported (Holst et al., 2010; Kramshøj et al., 2016). Different biogenic VOC may be related to pinenes from boreal forest (Tarvainen et

al., 2005) and sabinene from Siberian larches (Ruuskanen et al., 2007), while the snow-pack is a potential source of organic compounds (Grannas et al., 2007), and iodine oxide precursors (Raso et al., 2017). As particle growth rates at TIK are more rapid than other Arctic sites, it is probable that these terrestrial VOC sources play an important role. Particle mass loadings at TIK have also been shown to be especially high compared to other sites, and in the summertime, these are dominated by biogenic secondary aerosols (Moschos et al., 2022). The Alaskan continental UTQ site is most influenced by sea-ice related

air masses, with the CWT pointing towards the west as a strong source of particle growth driving vapour. This region has been

shown to be a driver of particle growth (Kolesar et al., 2017), and although the data were cleaned, an influence of anthropogenic gas emissions on the NPF at this site at unavoidable.

The back trajectory analyses performed here emphasise the influence of sea ice on NPF in the Arctic. Increased melting of sea ice in these regions, alongside thawing permafrost and precipitation changes related to warming will undoubtedly have profound effects on the NPF processes occurring. Prior long term analyses in the Arctic have shown regions of open water and melting sea ice to be related to NPF occurrence (Dall'Osto et al., 2017b, 2018b), In the Antarctic, melting sea ice is a source of amines in secondary aerosols (Dall'Osto et al., 2017a; Brean et al., 2021), and should sympagic conditions in the Arctic be similar, gas phase amines from sea ice melt will also accelerate particle formation rates by orders of magnitude. Further, increasing temperatures cause clear changes in continental emissions, such as the increases in biogenic emissions from tundra vegetation and changes in vegetation cover (Faubert et al., 2010; Peñuelas and Staudt, 2010; Potosnak et al., 2013; Lindwall et al., 2016). Increases to total aerosol surface area from increased sea spray due to sea ice melt may act as an efficient sink for low-volatility vapours, supressing future NPF (Browse et al., 2014). Taking all this into account, future Arctic melting can cause increases to emissions of multiple important new particle precursors, thus an acceleration of future Arctic NPF is possible. The complex interplay between source and sink of new aerosols must be understood in detail if the Arctic climate is to be predicted reliably in models.

**Conclusions**

Our results highlight the complex, multi-mechanistic system driving Arctic NPF. We show that particle formation and growth rates vary tremendously across the Arctic region, with vastly differing source regions producing vapour source rates spanning orders of magnitude in difference between sites. NPF frequency and intensity peak in the Arctic summer, with wintertime NPF being an infrequent phenomenon. Air masses from different Arctic eco-regions promote NPF at each of the sites (except those which are co-located), with gas-phase precursors from different source regions likely varying substantially, along with sources of organic and inorganic iodine and sulphur, as well as various organic compounds contributing to new particle formation, as shown by Beck et al., 2020 between Svalbard and the high Arctic. We present the first synchronous analysis of NPF at all of the longest-term Arctic aerosol measurement stations. While back trajectory analyses can point towards these source regions over long-terms, we still do not know the driving force behind NPF at these sites, as it is likely a combination of precursor emissions, photochemistry, ion pair production, temperature, and pre-existing surface area of aerosol. Measurements of particle size distributions down to critical cluster size and detailed chemical measurements are required to properly understand NPF at these sites.

## Acknowledgements

The aerosol and meteorological data for Utqiaġvik and Tiksi were downloaded from the International Arctic Systems for Observing the Atmosphere (www.iasoa. org) consortium website. For the Alert observations, we are grateful to the Canadian Department of National Defence, Andrew Platt, Sangeeta Sharma, Desiree Toom, Dan Veber and the Alert operators). Funding from the European Union's Horizon 2020 program grant agreement no. 654109 (ACTRIS) and INTAROS (project no. 727890) are acknowledged (E. Asmi). Observations at Zeppelin observatory were supported by Swedish Environmental Protection agency (Naturvårdsverket) and by ACAS project funded by Knut and Alice Wallenberg Foundation. The study was supported by UK Natural Environment Research Council (SEANA, NE/S00579X/1), the Spanish Ministry of Economy through project BIOeNUC (CGL2013−49020-R), PI-ICE (CTM2017−89117-R,) and the Ramon y Cajal fellowship (RYC-2012-11922). The authors also acknowledge financial support (to David C. S. Beddows) from the National Centre for Atmospheric Science (NCAS) (grant number R8/H12/83/011) funded by UK Natural Environment Research Council. Finally, the authors from Aarhus University were financially supported by the Danish Environmental Protection Agency and Danish Ministry for Climate Energy and Utilities via the MIKA/DANCEA funds for Environmental Support to the Arctic Region. As stressed in Freud et al. (2017), we would also like to express our appreciation and gratitude for the work and effort of all the scientists and engineers involved in setting up and maintaining the Arctic aerosol sites. Figures were created using the R software (R Core Team (2021) R: A Language and Environment for Statistical Computing. R Foundation for Statistical Computing, Vienna, Austria. https://www.R-project.org/).

## Additional information

Supplementary information is available in the online version of the paper.

## Author information

The authors declare no competing financial interests.

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

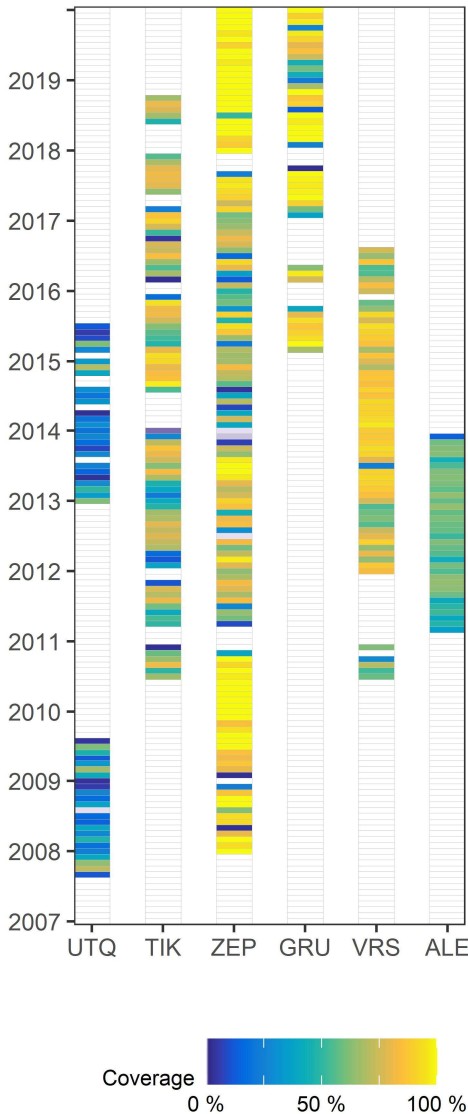

**Figure 1: Data coverage for each of the sites. Each individual cell corresponds to one full month of measurements. Fill colour corresponds to the total number of available hourly data as a percentage of the total hours within that month. The abbreviations along the bottom axis correspond to Utqiagvik, Tiksi, Mt. Zeppelin, Gruvebadet, Villum research station, and Alert.**

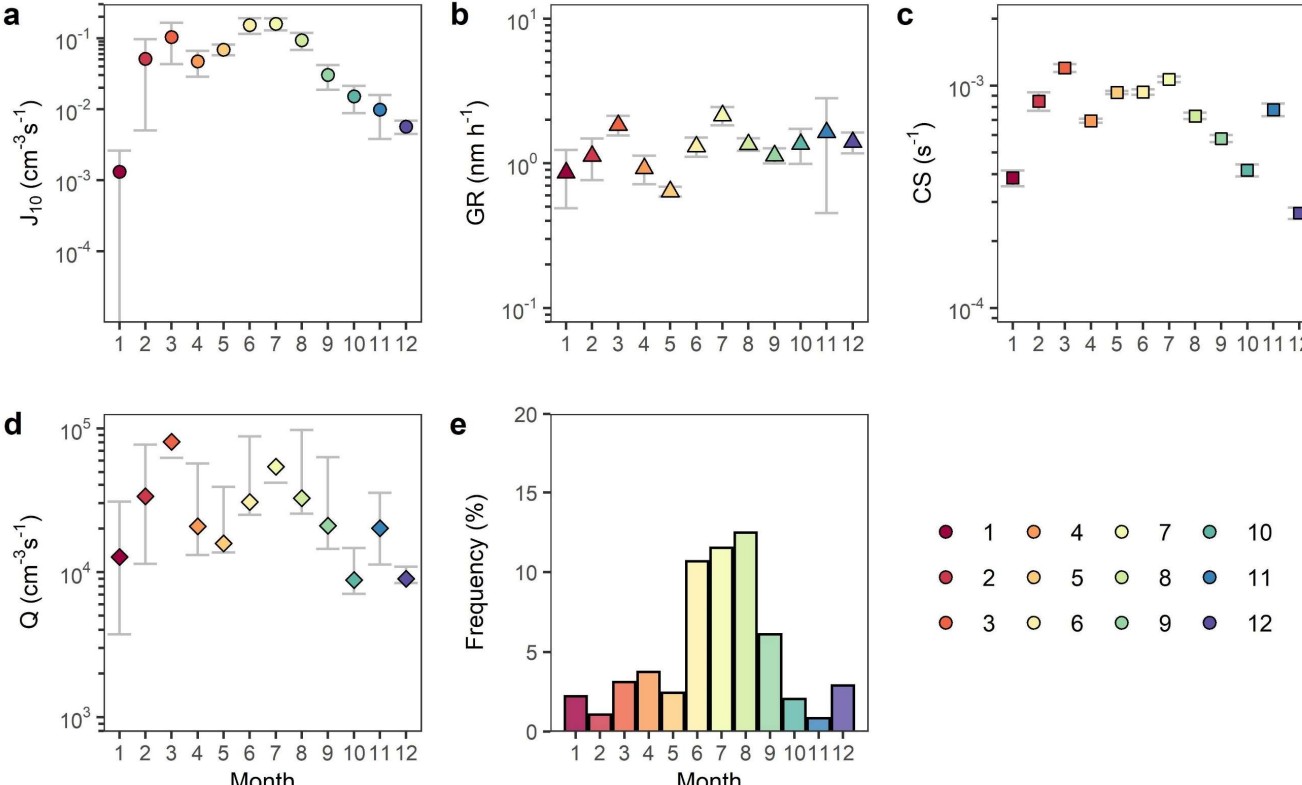

**Figure 2: Mean seasonal characteristics of NPF events from 6 Arctic sites, showing (a) formation rates at 10 nm, (b) growth rates, (c) condensation sinks during NPF events, (d) vapour source rates, and (e) NPF event frequency. Data points show the mean, error bars show one standard error on the mean. Data have been normalised to the size of the dataset relative to the average size of datasets, to avoid favouring datasets with longer runs of data. Colours represent each month.**

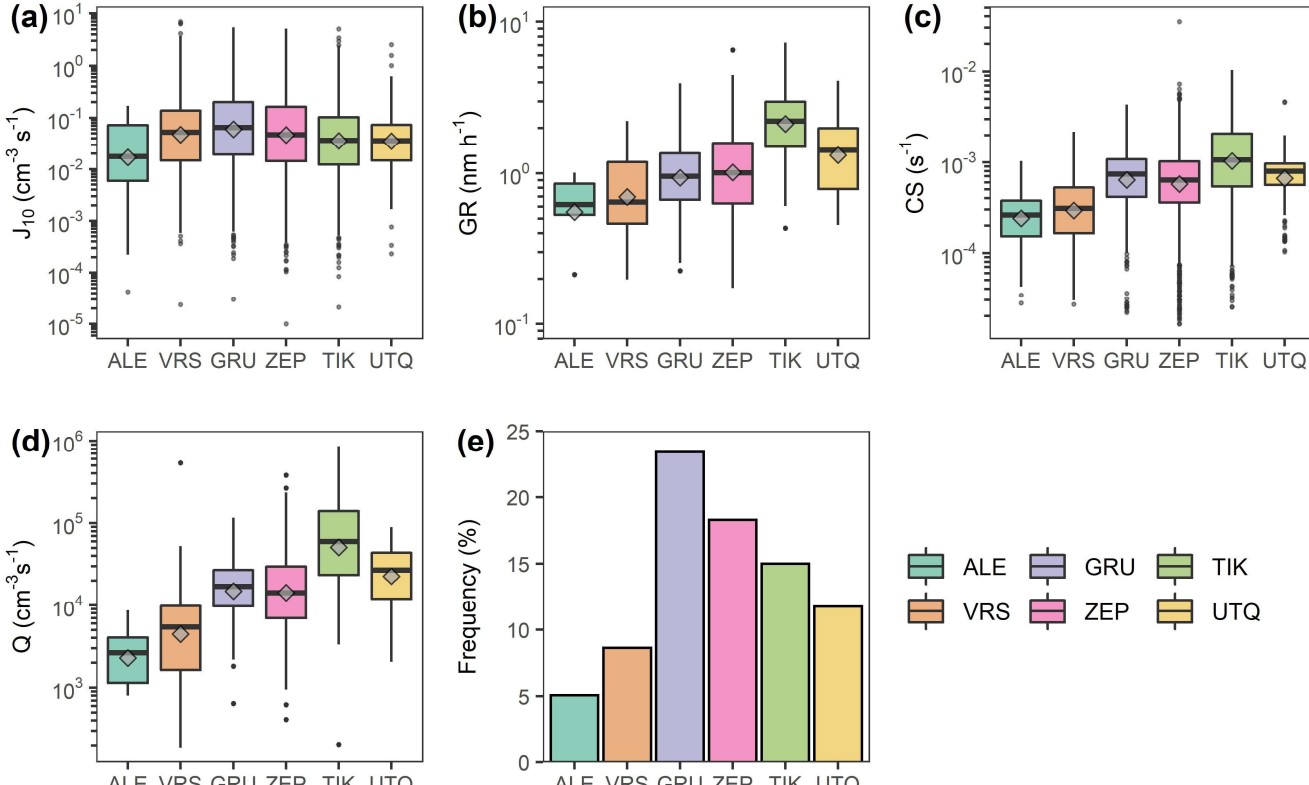

**Figure 3: Characteristics of NPF events per site in the months May through August inclusive, showing (a) formation rates at 10 nm, (b) growth rates, (c) condensation sinks, (d) vapour source rates, and (e) NPF event frequency. Box plots show median (centre line), mean (diamond), upper and lower quartiles (box limits), 1.5X interquartile range (whiskers), and any outliers as points. Colours represent each site.**

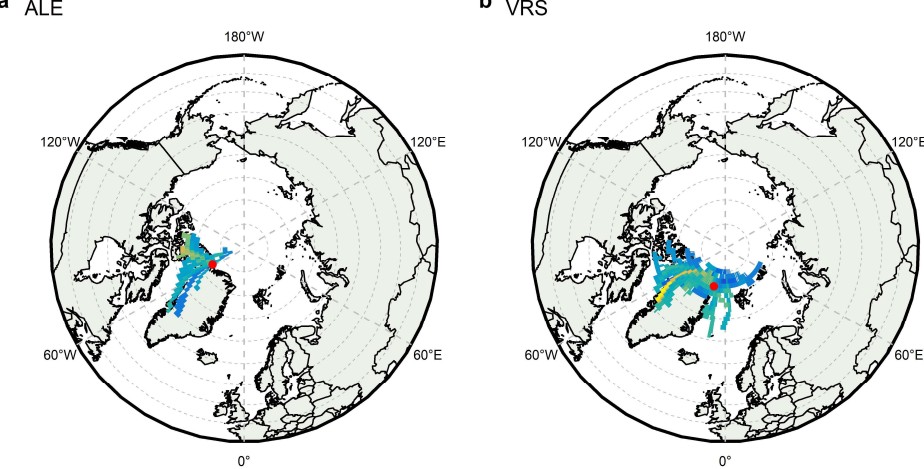

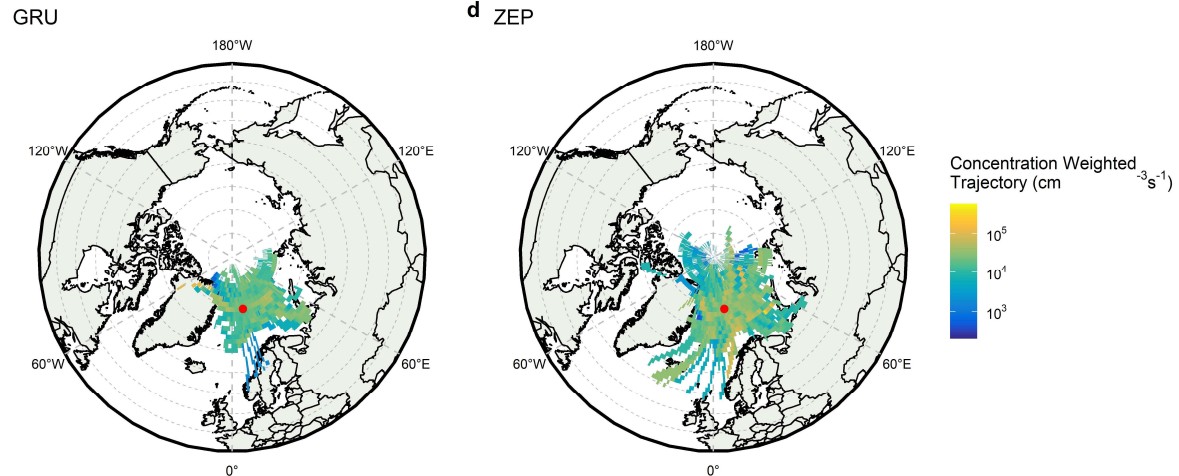

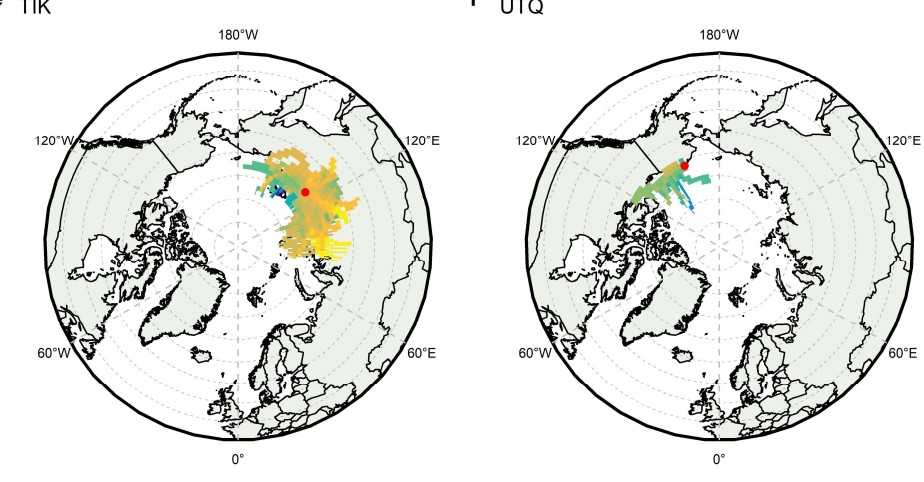

**Figure 4: 72-hour back trajectories plotted on a 1º by 1º grid, weighted by the source rate of equivalent sulphuric acid vapour driving particle growth, for (a) ALE, (b) VRS, (c) GRU, (d) ZEP, (e) TIK, and (f) UTQ.**

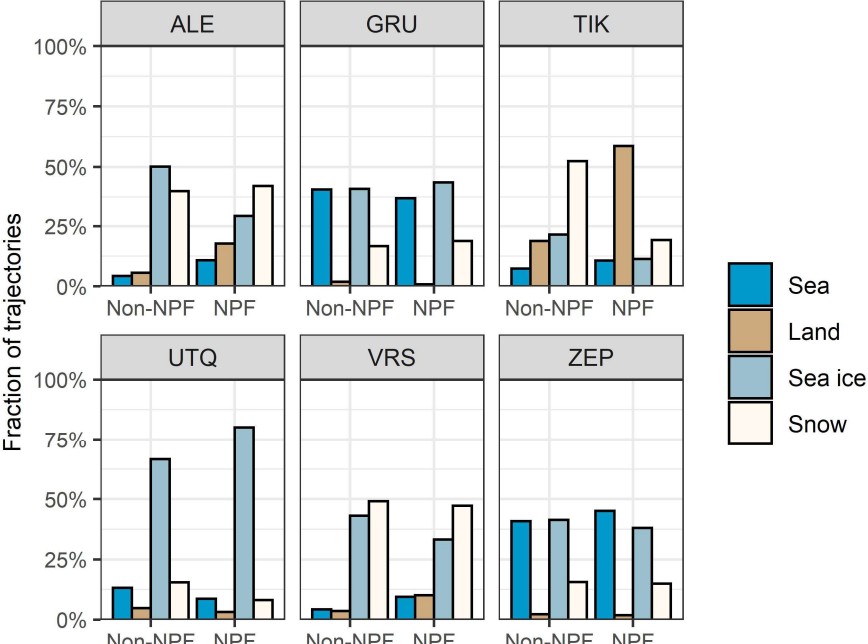

**Figure 5: Link between surface type and 72-hour HYSPLIT back trajectory points during NPF events, and outside of NPF events. Air masses were assigned a surface type based upon a 24 km grid of IMS data.**

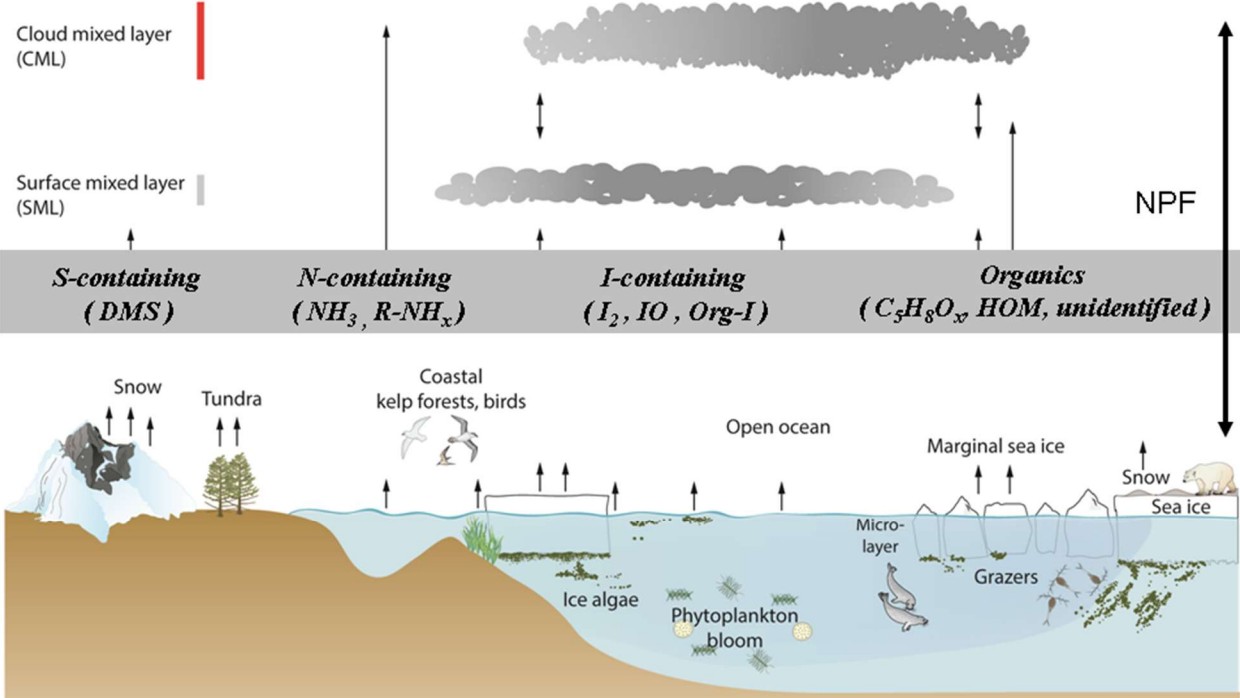

**Figure 6: Schematic illustrations of the sea-ice, microbiota, sea-to-air emissions and New Particle Formation (NPF) occurring in the typical summertime stratus-topped Arctic boundary layer. Vertical red and grey bars broadly indicate Cloud Mixed Layer (CML) and Surface Mixed Layer (SML) as inspired by Brooks et al. (2017). The grey box indicates known possible gas-phase NPF precursors from the potential Arctic natural terrestrial and marine sources drawn below**