# Peer review of "Collective geographical eco-regions and precursor sources driving Arctic new particle formation"

_Atmospheric Chemistry and Physics, 2022_

## Referee Comment (RC2)

**Review_acp-2022-280: James Brean et. al.,**

Collective geographical eco-regions and precursor sources driving Arctic new particle formation

**General comments:** This manuscript analyzes atmospheric particle formation and growth rates for six Arctic sites.

The manuscript provides some useful scientific contribution associated with new particle formation (NPF) in the Arctic. It is a similar idea like the paper from Sellegri et. al., 2019 [Atmosphere 2019, 10, 493; doi:10.3390/atmos10090493] for different high mountain research stations. This paper was not cited. a reference to this paper would have been very helpful here and also a short introduction of the theory to explain the key processes. This is totally missing. The manuscript concentrates too much on statistics and source area and the information on the specific aspects for the different locations are more or less missing. This is much better explained by the Sellegri paper. Some changes in the structure and presentation would be nice. The following questions/concerns should be satisfactorily addressed prior to final publication.

**Specific comments**

Figure 1: show the general seasonality of the different parameters: J10, GR, CS and Q. here is missing the data availability of the different stations – is the time period the same and the amount of data's for the different stations. This figure is too general and only J10 and Frequency have the same signature.

Figure 2: Here is the information, separated for the six stations and also not really clear message. Different values for the parameter at different stations. It would be better, to use Figure 1 to say the NPF Frequency is highest in June/July/August and present in Figure 2 the data sets only for these three months. Then the differences between the station can be better explained….

Figure 3: a specific inside view in the source area seems better instead of this extreme general picture over the entire data set. For me is here Figure S5 much better and Figure 3 should remove for Figure S5.

Figure 4: is very clear and good described in the discussion (starting L253)

Figure 5: I don't see a big motivation for this figure, could be remove

Figure S1: this is mandatory in the manuscript, and not in the supplementary! But you see the limited data basis and also the lack of data from different station for different time period. This long measurement period from Zeppelin to compare with other station with very limited time span seems critical and should be better discussed in the text. Only 2015 show an observation overlap from 5 station. A specific discussion of this time period is here recommended. Is the result from this period similar / same for the entire period?

Figure S3: why as example only type 1,2,3 for Tiksi – is that a typical signature or a special in compare to the stations?

L66: It is not enough to write, NPF is a deeply complex process – a little bit more on the theory and main processes would be very helpful

L81: It is here missing to mention, what are the key parameter for NPF in the Arctic. Are the same like in other regions or not.

L82: The sentence should end with dot, that is missing

L89-105: The table S1 on the station is not complete, the used instruments is here also recommended, including the specific information, whether the different systems at the different stations means special constrains for the data analysis. How big are the differences between TSI 3034, TSI 3772 CPC and twin DMPS, custom built SMPS and TSI 3010 CPC. This could show very easy in a table…

L114: Is the condensation sinks CS the main parameter identity NPF or the particle growth GR. I think the formulas are for this manuscript secondary. The list of priority for the Arctic site seems more attractive.

L143/144: The explanation, why only type A and type B are used for the identification of NPF, is missing.

L184: the discussion of the spatial variability is in general attractive, but the information in the following lines up to 245 is very unstructured. Here a table with the different mean parameter for the three regions makes the discussion on the differences and explanation of reason much easier.

L249/250: I see also a variation of CS at the different sites and a focus to the high frequency period of the NPF could be bring a better inside view.

L253-255: what is the motivation for Figure 5 and this sentence? Please explain it.

L303-311: Here the message is not clear enough. Figure 1 for example show a high seasonality and for the different parameter not a complete peak for summer, sometime also NPF events in winter. Too much statistics is here not perfect. A specific final statement, what is the key parameter for the NPF event in the Arctic and what are the difference between the three locations would be helpful.

---

## Author Comment (AC1)

**Collective geographical eco-regions and precursor sources driving Arctic new particle formation, J. Brean et al. Responses to RC1.**

Note: Figures in the manuscript and SI are referred to here as *"Figure X"*, figures included in these review responses, but not in the manuscript and SI are referred to as *"Response Figure Y"*. Review comments are displayed in blue, and sections that have been added to the text are coloured *green*. We thank the reviewer for their insightful comments and provide responses below.

**General comments:** This manuscript analyzes atmospheric particle formation and growth rates for six Arctic sites. Back trajectories are used to examine the potential locations for particle precursors. The manuscript provides a useful scientific contribution since there remains considerable uncertainty about sources and processes associated with new particle formation (NPF)/growth in the Arctic. The following questions/concerns should be satisfactorily addressed prior to final publication.

**Specific comments:**

Figure 1: Please consider adding numerical labelling to the vertical axis for GR, CS, and Q. Currently there is only 1 tick labelled for the GR, CS, and Q panels.

These have been updated as below, in lieu of adding numbers at every tick (as there would be little room for it), we have included more numbers at either ends of the axes.

[Figure]

**Figure 2: Mean seasonal characteristics of NPF events from 6 Arctic sites, showing (a) formation rates at 10 nm, (b) growth rates, (c) condensation sinks during NPF events, (d) vapour source rates, and (e) NPF event frequency. Data points show the mean, error bars show one standard error on the mean. Data have been normalised to the size of the dataset relative to the average size of datasets, to avoid favouring datasets with longer runs of data. Colours represent each month.**

Figure 4: To help with the interpretation of Fig. 4, which of these differences are statistically significant? For example, for GRU and ZEP, there does not appear to be any appreciable difference between NPF and non-NPF trajectories. However, for ALE, TIK and VRS, there are increases in the trajectory fraction over sea and land for NPF versus non-NPF, but it is uncertain if these differences

are statistically significant. Likewise for UTQ, there is an increase in time over sea ice for NPF versus non-NPF, but is this a statistically significant difference?

These percentages are not averages, they are the sum of the total number of hours travelling over different regions (ocean, sea ice, land, snow) as indicated "between surface type and 72-hour HYSPLIT back trajectory points during NPF events, and outside of NPF events". Therefore, they should be interpreted as they are and compared among the sites, we highlight through the text when these differences are small or large.

Figure 5: The schematic is helpful to visually indicate the variety of sources. There are aspects of the figure that are somewhat confusing and could be modified for a cleaner presentation. What do the vertical red and grey bars near CML and SML indicate – are they needed? What is the meaning of the large grey box – is it needed? Could the number of arrows on the figure be reduced? Why are some arrows longer than others? The grey box appears to list the precursors, but it is unclear if they are meant to be vertically aligned over certain sources.

Among all co-authors, we spent a fair amount of time drawing the Figure 5 (now Figure 6). Answering questions editing in the figure caption. We apologize we forgot to cite Brooks et al 2017, now edited. The caption now reads as follows

"Figure 6: Schematic illustrations of the sea-ice, microbiota, sea-to-air emissions and New Particle Formation (NPF) occurring in the typical summertime stratus-topped Arctic boundary layer. Vertical red and grey bars broadly indicate Cloud Mixed Layer (CML) and Surface Mixed Layer (SML) as inspired by Brooks et al. (2017). The grey box indicates known possible NPF gas precursor from potential Arctic natural terrestrial and marine sources drown below."

Brooks, I. M., Tjernström, M., Persson, P. O. G., Shupe, M. D., Atkinson, R. A., Brooks, B. J. (2017). The turbulent structure of the Arctic summer boundary layer during The Arctic Summer Cloud-Ocean Study. Journal of Geophysical Research: Atmospheres, 122, 9685–9704, https://doi.org/10.1002/2017JD027234

Table S1: Please replace BAR with UTQ for consistency with the rest of the manuscript.

Thank you for pointing this out. We have amended this.

Figure S1: The meaning of the faint lines is unclear.

Each of these lines represented the border between each month. Some lines were lost due to compression upon saving. We provide the figure again below and update it in the manuscript. This has also been moved to be our Figure 1 after a suggestion from another reviewer.

[Figure]

Figure 1: Data coverage for each of the sites. Each individual cell corresponds to one full month of measurements. Fill colour corresponds to the total number of available hourly data as a percentage of the total hours within that month.

Figure S4: Is the hour of day in local time?

Yes. The figure caption has been updated to include this.

L26: The authors comment that the mean frequency and particle formation rates vary greatly between sites. How much of this variability can be attributed to differences in the years that measurements were available for the various sites?

We did indeed check this in the early stages of data analysis. Variations in the key parameters we discuss at most sites vary around what is normal for year-by-year variations in NPF, with no overall trends upwards or downwards. The growth rates at barrow show a sharp upward trend in the final two years of measurement, but the total number of events measured is relatively low. There seems to be no overall trend in condensation sinks during NPF.

[Figure]

L42: How do the predicted and observed temporal trends for cloud cover differ?

Good question! We include the following line in the paper (bold)

"However, atmospheric chemical transport and climate models consistently fail to replicate much of the observed variation of aerosol concentrations observed at ground-based stations (Sand et al., 2017) and recently there has been shown a different temporal trend in predicted and observed cloud cover at a high Arctic site (Gryning et al. 2021) **continuing a historical overestimation of low-level clouds in the Arctic by climate models, particularly in the wintertime, most particularly in daily data (Quan et al., 2012).**

Qian, Y., Long, C. N., Wang, H., Comstock, J. M., McFarlane, S. A., and Xie, S.: Evaluation of cloud fraction and its radiative effect simulated by IPCC AR4 global models against ARM surface observations, Atmos. Chem. Phys., 12, 1785–1810, https://doi.org/10.5194/acp-12-1785-2012, 2012."

L47: The authors draw focus to neglect of iodine nucleation in models – are there other mechanisms that are also often neglected as related to organics etc.?

Yes, indeed particle formation from oxygenated organic molecules is often neglected in these models. In Gordon et al. (2017), parametrisations from the CLOUD chamber studies of particle formation from a host of compounds are included. We update this sentence to highlight the specificity to this study, and also reference some recent work highlighting the importance of an additional mechanism in the upper troposphere

"NPF is estimated to be responsible for around half of global cloud condensation nuclei (CCN) concentrations **when neutral and ion-induced mechanisms of particle formation from sulphuric acid, ammonia, and organics are considered** (Gordon et al., 2017). These models neglect mechanisms such as iodine nucleation shown to be important in the high Arctic (Baccarini et al., 2020), **amines as stabilisers, also important in Antarctica (Brean et al., 2021), and particle formation involving nitric acid, important in the upper troposphere (Wang et al., 2022). Other models neglect all mechanisms except those involving sulphuric acid (Liu et al., 2012).**

**Liu, X., Easter, R. C., Ghan, S. J., Zaveri, R., Rasch, P., Shi, X., Lamarque, J.-F., Gettelman, A., Morrison, H., Vitt, F., Conley, A., Park, S., Neale, R., Hannay, C., Ekman, A. M. L., Hess, P., Mahowald, N., Collins, W., Iacono, M. J., Bretherton, C. S., Flanner, M. G., and Mitchell, D.: Toward a minimal representation of aerosols in climate models: description and evaluation in**

the Community Atmosphere Model CAM5, Geosci. Model Dev., 5, 709–739, https://doi.org/10.5194/gmd-5-709-2012, 2012.

Wang, M., Xiao, M., Bertozzi, B., Marie, G., Rörup, B., Schulze, B., Bardakov, R., He, X.-C., Shen, J., Scholz, W., Marten, R., Dada, L., Baalbaki, R., Lopez, B., Lamkaddam, H., Manninen, H. E., Amorim, A., Ataei, F., Bogert, P., Brasseur, Z., Caudillo, L., De Menezes, L.-P., Duplissy, J., Ekman, A. M. L., Finkenzeller, H., Carracedo, L. G., Granzin, M., Guida, R., Heinritzi, M., Hofbauer, V., Höhler, K., Korhonen, K., Krechmer, J. E., Kürten, A., Lehtipalo, K., Mahfouz, N. G. A., Makhmutov, V., Massabò, D., Mathot, S., Mauldin, R. L., Mentler, B., Müller, T., Onnela, A., Petäjä, T., Philippov, M., Piedehierro, A. A., Pozzer, A., Ranjithkumar, A., Schervish, M., Schobesberger, S., Simon, M., Stozhkov, Y., Tomé, A., Umo, N. S., Vogel, F., Wagner, R., Wang, D. S., Weber, S. K., Welti, A., Wu, Y., Zauner-Wieczorek, M., Sipilä, M., Winkler, P. M., Hansel, A., Baltensperger, U., Kulmala, M., Flagan, R. C., Curtius, J., Riipinen, I., Gordon, H., Lelieveld, J., El-Haddad, I., Volkamer, R., Worsnop, D. R., Christoudias, T., Kirkby, J., Möhler, O., and Donahue, N. M.: Synergistic HNO3–H2SO4–NH3 upper tropospheric particle formation, Nature, 605, 483–489, https://doi.org/10.1038/s41586-022-04605-4, 2022."

L68: At what spatial scale are the eco-regions defined? Please clarify here about which eco-regions will be employed for this analysis.

Eco-regions are not defined in a specific spatial scale. In spatial ecology, scale refers to the spatial extent of ecological processes and the spatial interpretation and availability of the data. As written in the manuscript

"Broadly, here we define an eco-region as an ecologically and geographically defined area that captures not only the distribution of biological communities but also the environmental conditions (including climate variables) such as ice sheet, marginal sea-ice zone, tundra, snow-covered land, sea-ice influenced open ocean, permanent open ocean, animal-colonised shores and islands, etc. (Barry et al., 2013; Meltorft et al., 2013; CAFÉ 2017; Schmale et al., 2021). Research at different sites has, for example, pointed towards sea ice (Allan et al, 2015; Baccarini et al, 2020; Dall'osto et al., 2017b; Dall'osto et al., 2018b; Heintzenberg et al., 2015), and open water (Dall'osto et al., 2018b; Croft et al., 2019; Wiedensohler et al., 1996; Willis et al., 2017) regions as sources of new particle precursors"

In this analysis, we can only refer to land, sea ice, ocean and snow as geographically defined regions (satellite data). However, when discussing the possible NPF precursors, we discuss potential eco-regions including different biotic and abiotic processes from natural terrestrial and marine sources.

L78: There is mention of a lack of 'simultaneous' comparisons – 'simultaneous' seems redundant here.

We agree, and have amended the text accordingly

L110: How reliable are the measurements between 10 and 20 nm for the various instruments?

We have no reliable estimates of the uncertainties for each individual instrument, but these will depend on both differences between instruments, and the corrections performed on the data (CPC counting efficiency, DMA transfer function, pipe losses etc.) which are especially pertinent for the smallest size fractions. Previous intercomparison works show that different inversion routines account for some few percent difference in the size distribution, while differences between instruments from different manufacturers are within 10% difference for the 20 – 200 nm size range (Wiedensohler et al., 2012). We provide a discussion of such errors in the text as below

"Intercomparison workshops have shown differences between instruments measuring particle size distributions to be within 10%, increasing at smaller diameters (Wiedensohler et al., 2012). This

produces some uncertainty when we are comparing particle formation rates and growth rates of particles in these smaller size regimes, but this uncertainty is substantially smaller than the differences in particle concentrations between sites."

Wiedensohler, A., Birmili, W., Nowak, A., Sonntag, A., Weinhold, K., Merkel, M., Wehner, B., Tuch, T., Pfeifer, S., Fiebig, M., Fjäraa, A. M., Asmi, E., Sellegri, K., Depuy, R., Venzac, H., Villani, P., Laj, P., Aalto, P., Ogren, J. A., Swietlicki, E., Williams, P., Roldin, P., Quincey, P., Hüglin, C., Fierz-Schmidhauser, R., Gysel, M., Weingartner, E., Riccobono, F., Santos, S., Grüning, C., Faloon, K., Beddows, D., Harrison, R., Monahan, C., Jennings, S. G., O'Dowd, C. D., Marinoni, A., Horn, H.-G., Keck, L., Jiang, J., Scheckman, J., McMurry, P. H., Deng, Z., Zhao, C. S., Moerman, M., Henzing, B., de Leeuw, G., Löschau, G., and Bastian, S.: Mobility particle size spectrometers: harmonization of technical standards and data structure to facilitate high quality long-term observations of atmospheric particle number size distributions, Atmos. Meas. Tech., 5, 657–685, https://doi.org/10.5194/amt-5-657-2012, 2012

L122: How is CoagS_dp calculated?

CoagS$_{dp}$ is calculated from the collision frequency function (equation shown below). We include a reference to a paper explaining this calculation in the text

$$CoagS_{dp} = \int_{dp0}^{\infty} \beta_{dp,dp0} N_{dp} d dp$$

Eq. (4): What is the definition of lambda? How could these calculations and conclusions differ if the particle growth was driven by vapors other than sulphuric acid?

This is a good point that requires clarification in the text. We include a missing reference here to Nieminen et al. (2010) and update the text to explain that lambda is the mean free path of condensing vapour. Our conclusions would be the same regardless of the values chosen for density, mass, and diffusivity as we do not draw any conclusion from the absolute values, but from their relative values.

Nieminen, T., Lehtinen, K. E. J., and Kulmala, M.: Sub-10 nm particle growth by vapor condensation – effects of vapor molecule size and particle thermal speed, Atmos. Chem. Phys., 10, 9773–9779, https://doi.org/10.5194/acp-10-9773-2010, 2010.

The residence time in each cell is used in the calculation of the concentration-weighted trajectory – does this assume that at all times, the vapor source rate will be the same for all grid boxes? Please clarify and if so, how does this assumption impact the conclusions?

For each trajectory (covering many grid squares) there is indeed a singular $Q$ value. This is a problem when a small number of trajectories are present, or when there are strong point sources for emissions. In our case, we have a high number of trajectories, and broad source regions, so this does not lead to any changes in conclusions.

Section 2.4: 3-day back trajectories are used to examine the regions that the air mass has passed over prior to arriving at the time of NPF. To help with interpreting these trajectories, please indicate the expected lifetime of the precursor vapors. How do vapor aging processes impact this calculation?

The answer to this depends on the vapour in question, the answer to which we do not know. The lifetime of DMS with regards to oxidation is fast. The subsequent oxidation of SO$_2$ to H$_2$SO$_4$ is then slow, giving SO$_2$ a rather longer lifetime much greater than 72 hours (von Glasow et al., 2009). The oxidation of most organics is also fast (i.e., many monoterpenes and isoprene (Fuentes et al., 2000) have lifetimes of hours). The formation of highly oxygenated molecules is then typically also fast, driven by autoxidation. Once oxygenated, these compounds' lifetimes with respect to gas-phase chemical reactions tend to be much lower than their lifetime with respect to condensation, on the

order of $10^1 - 10^2$ mins for these Arctic CS values (Bianchi et al., 2019). The lifetime of iodic acid precursors such as hypoiodous acid are likely also low (Sherwen et al., 2016). 72 hours was chosen as somewhere between the longer lifetime of $SO_2$ and shorter lifetimes of organics and iodine compounds. In the text we include the following

"72 hours lies somewhere between the long atmospheric lifetime of $SO_2$ (von Glasow et al., 2009), and the shorter lifetime of reactive VOCs, MSA, and iodine compounds (Fuentes et al., 2000; Sherwen et al., 2016; Kloster et al., 2006)"

Bianchi, F., Kurtén, T., Riva, M., Mohr, C., Rissanen, M. P., Roldin, P., Berndt, T., Crounse, J. D., Wennberg, P. O., Mentel, T. F., Wildt, J., Junninen, H., Jokinen, T., Kulmala, M., Worsnop, D. R., Thornton, J. A., Donahue, N., Kjaergaard, H. G., and Ehn, M.: Highly Oxygenated Organic Molecules (HOM) from Gas-Phase Autoxidation Involving Peroxy Radicals: A Key Contributor to Atmospheric Aerosol, Chem. Rev., 119, 3472–3509, https://doi.org/10.1021/acs.chemrev.8b00395, 2019.

Fuentes, J. D., Lerdau, M., Atkinson, R., Baldocchi, D., Bottenheim, J. W., Ciccioli, P., Lamb, B., Geron, C., Gu, L., Guenther, A., Sharkey, T. D., and Stockwell, W.: Biogenic Hydrocarbons in the Atmospheric Boundary Layer: A Review, Bull. Am. Meteorol. Soc., 81, 1537–1575, https://doi.org/10.1175/1520-0477(2000)081<1537:BHITAB>2.3.CO;2, 2000.

Kloster, S., Feichter, J., Maier-Reimer, E., Six, K. D., Stier, P., and Wetzel, P.: DMS cycle in the marine ocean-atmosphere system – a global model study, Biogeosciences, 3, 29–51, https://doi.org/10.5194/bg-3-29-2006, 2006.

von Glasow, R., Bobrowski, N., and Kern, C.: The effects of volcanic eruptions on atmospheric chemistry, Chem. Geol., 263, 131–142, https://doi.org/10.1016/j.chemgeo.2008.08.020, 2009.

Sherwen, T., Evans, M. J., Carpenter, L. J., Andrews, S. J., Lidster, R. T., Dix, B., Koenig, T. K., Sinreich, R., Ortega, I., Volkamer, R., Saiz-Lopez, A., Prados-Roman, C., Mahajan, A. S., and Ordóñez, C.: Iodine's impact on tropospheric oxidants: A global model study in GEOS-Chem, Atmos. Chem. Phys., 16, 1161–1186, https://doi.org/10.5194/acp-16-1161-2016, 2016.

L151: How does the 1x1 degree resolution of the grid cells impact the results?

This resolution degree is commonly used in trajectory analyses (i.e., Fleming et al., 2012). As the emissions we are observing occur across large geographical ranges (i.e., open ocean, sea ice) which tends to extend across many grid squares, an increase in resolution would only increase our uncertainty without providing useful information, as the uncertainty existing in the HYSPLIT model is quoted on the NOAA website to be "15 to 30% of the travel distance"

Fleming, Z. L., Monks, P. S., and Manning, A. J.: Review: Untangling the influence of air-mass history in interpreting observed atmospheric composition, Atmos. Res., 104–105, 1–39, https://doi.org/10.1016/j.atmosres.2011.09.009, 2012.

L157: How does neglect of trajectories above 1 km impact the conclusions?

We neglect approximately 2.2% of the trajectories. The frequency of these trajectories is plotted below. Most of these exclusions are at ZEP, with very few at TIK or UTQ. As the total number of exclusions is small, the effect on data is minimal. We include the following line in the paper

"Trajectories more than 1,000 m a.g.l. were not considered in these analyses, **excluding 2.2% of trajectories, mostly at ZEP**."

[Figure]

Response Figure 1: Map of excluded trajectories of heights >1000 m, gridded to a 1x1 degree grid

L167: Why does the condensation sink not appear greater in the Arctic Haze season? This seems unexpected – what contributes to this lack of difference between seasons?

 We show the CS during NPF periods. This wintertime NPF is infrequent, and occurs under periods of lower CS than the seasonal average. We update the text to properly reflect this

While winter and springtime periods are typically associated with higher accumulation mode loading due to Arctic Haze (Abbatt et al., 2019; Asmi et al., 2016; Heintzenberg et al., 2017), no significant difference is seen in CS between seasons **during NPF periods**, although individual months vary by over a factor of 3 (Fig. 1). CS between NPF and non-NPF events **across the whole year** is also similar (mean CS $8.6 \cdot 10^{-4}$ s$^{-1}$ and $8.9 \cdot 10^{-4}$ s$^{-1}$ during NPF and non-NPF periods, respectively). **It is worth note that wintertime NPF, although making up a small number of total NPF events, tends to occur under a 15% lower mean CS than the seasonal average.**

L174: Is this local time?

 Yes, all times are in local time. We provide clarification for this above

"Throughout this text, the seasons are defined as spring (MAM), summer (JJA), autumn (SON) and winter (DJF). **All times are in local time**."

L209: What is the driver of NPF at VRS if there appears to be no link to any specific eco-region?

There are plenty of possible NPF drivers of NPF that we may not capture here. Precursor emission rates can of course vary within a single region, different oxidation rates of those precursors to sulphuric acid, HOMs, iodic acid etc., and temperature dependent effects, none of which are within the scope of this work.

L220-221: Please check the wording here – it is not clear that the concentration-weighted trajectories can conclusively show that vapours driving particle growth "come from all the surrounding open ocean and sea ice regions". Certainly, there is the potential for contribution from any of these regions – but is it possible that some regions might contribute more strongly than others in a manner that is not considered by this analysis approach?

Very correct! We amend this wording as below

"the CWTs for Q **point to potential source regions of precursor vapours for particle growth from**  all the surrounding open ocean and sea ice regions"

L234: 'strongest vapour source' – do you mean '…source region'?

 Yes. We also change this wording

"The CWT and land type analysis indicates that the  **source region most strongly associated with high values of Q** at TIK"

L239: Should this association of the sea ice with the oil fields be indicated in caption of Fig. 4 to indicate that sea ice regions and oil-field regions could not be separated?

We do not mean to imply that we cannot separate these, we just mean to state that the trajectories are heavily sea-ice influenced, while also travelling around regions where oil-field regions are to be found.

L242-245: Consider merging this 1-sentence paragraph with related text elsewhere.

Thank you. Done, this was an error.

L252: 'calculated values are similar' – where is this shown?

We opt not to directly quote these numbers in our paper, but update this section to read as follows

"The site-by-site variation **in CS, J, and Q were tabulated** recently in a review paper by Schmale and Baccarini (2021), to which our calculated values are similar where similar numbers are available **(Figures 1 and 2). For the sites where figures are not available, we provide the first reports of key NPF parameters**"

L253: 'strong source regions' – please indicate what is meant by 'strong' – is there a certain magnitude?

"We show that the vapours which drive particle growth each of these sites often, but not always  **coincide with air masses flowing over particular, directional source regions** (Fig. 3)."

L261: Are there any other potential reasons for slow particle formations rates, in addition to iodine oxoacid related NPF?

Yes. In the introduction we state

"NPF events are dependent upon the precursor vapour concentrations, temperature, ion pair production rate, and the surface area of pre-existing aerosols, and thus this CCN contribution varies regionally, and is a result of an interplay between these factors (Lee et al., 2019)."

We add the following sentence in the discussion of VRS and ALE

**"If iodine oxoacids are not the species responsible for particle formation observed in this dataset, these low formation rates may be related to low concentrations of alternative precursors, and weak solar radiation reducing both rates of photochemistry and ion pair production."**

L266: The potential vapour source from the coast of Greenland is interesting – could there be other sources in addition to DMS here?

Yes, we neglect to include seabird-related NH₃ which is substantial here

"Source apportionment studies applied to highly time resolved VOC data show coastal Greenland to be a dominant source of DMS (Pernov et al., 2021), **as well as ammonia from seabird colonies (Riddick et al., 2012).**"

L290: Regarding the influence of sea ice on NPF – what is the potential of vapors from open leads in the sea ice to also make a contribution?

There is a likely contribution from open leads. This has been observed in the case of dimethylsulphide (i.e., Levasseur, 2013), and long-term studies have hypothesised contributions from these regions (i.e., Dall'Osto et al., 2018). We include a mention of this in the main text.

Dall'Osto, M. ;, Simo, R. ;, Harrison, R. ;, Beddows, D. ;, Saiz-Lopez, A. ;, Lange, R. ;, Skov, H. ;, Nøjgaard, J. K. ;, Nielsen, I. E. ;, and Massling, A.: Abiotic and biotic sources influencing spring new particle formation in North East Greenland, Atmos. Environ., 190, 126–134, https://doi.org/10.1016/j.atmosenv.2018.07.019, 2018.

Levasseur, M.: Impact of Arctic meltdown on the microbial cycling of sulphur, Nat. Geosci., 6, 691–700, https://doi.org/10.1038/ngeo1910, 2013.

L290: The sentence starting with 'Increased melting of permafrost…" seems not to fit in this discussion on sea ice – would text this fit better elsewhere?

This sentence has been edited to make this point clearer

"The back trajectory analyses performed here emphasise the influence of sea ice on NPF in the Arctic. **Increased melting of sea ice in these regions, alongside** permafrost and precipitation related to warming will undoubtedly have profound effects on the NPF processes occurring"

L307: 'likely varying wildly' – are there references that could support this speculation?

We agree "wildly" was an odd choice of wording here. We amend as follows

"Air masses from different Arctic eco-regions promote NPF at each of the sites (except those which are co-located), with gas-phase precursors from different source regions likely varying  **substantially**, with sources of organic and inorganic iodine and sulphur, as well as various organic compounds contributing to new particle formation, **as shown by Beck et al., 2020 between Svalbard and the high Arctic"**

---

## Author Comment (AC2)

**Collective geographical eco-regions and precursor sources driving Arctic new particle formation, J. Brean et al. Responses to RC1.**

Note: Figures in the manuscript and SI are referred to here as *"Figure X"*, figures included in these review responses, but not in the manuscript and SI are referred to as *"Response Figure Y"*. Review comments are displayed in blue, and sections that have been added to the text are coloured *green*. We thank the reviewer for their insightful comments and provide responses below.

**General comments**: This manuscript analyzes atmospheric particle formation and growth rates for six Arctic sites.

The manuscript provides some useful scientific contribution associated with new particle formation (NPF) in the Arctic. It is a similar idea like the paper from Sellegri et. al., 2019 [Atmosphere 2019, 10, 493; doi:10.3390/atmos10090493] for different high mountain research stations. This paper was not cited. a reference to this paper would have been very helpful here and also a short introduction of the theory to explain the key processes. This is totally missing. The manuscript concentrates too much on statistics and source area and the information on the specific aspects for the different locations are more or less missing. This is much better explained by the Sellegri paper. Some changes in the structure and presentation would be nice. The following questions/concerns should be satisfactorily addressed prior to final publication.

Thank you for providing these comments. Alongside addressing the specific comments below, we include a reference to the highly useful Sellegri paper in the manuscript in the following sentence

"Motivated by the lack of studies comparing NPF at these sites simultaneously, **and following in the stead of previous publications studying multiple sites in different environments (e.g., Sellegri et al., 2019)** with calculations of the key parameters of particle formation…"

**Specific comments:**

Figure 1: show the general seasonality of the different parameters: J10, GR, CS and Q. here is missing the data availability of the different stations – is the time period the same and the amount of data's for the different stations. This figure is too general and only J10 and Frequency have the same signature.

We include this figure to indicate seasonal variations in NPF frequency and formation rate, which leads firmly into Figure 2. We would not necessarily expect the same signature to be seen in the other features (GRs, CSs etc.) and the lack of a summertime maximum in growth rate across the Arctic is a useful finding. As the data availability is not included, we now include our data availability as Figure 1 (see below).

Figure 2: Here is the information, separated for the six stations and also not really clear message. Different values for the parameter at different stations. It would be better, to use Figure 1 to say the NPF Frequency is highest in June/July/August and present in Figure 2 the data sets only for these three months. Then the differences between the station can be better explained….

This is exactly what this figure is. The figure caption states this explicitly

"Figure 2: Characteristics of NPF events per site in the months May through August inclusive, showing…"

The title to section 3.2 also indicates this

**"3.2 Spatial variation of summertime NPF features"**

Which begins:

"The site-by-site variation in summertime NPF event characteristics is shown in Fig. 3…"

To make this clearer, we update the abstract as follows

"…and particle formation rates themselves vary greatly between sites, highest at Svalbard, and lowest in the high Arctic. **Summertime** growth rate, condensational sinks…"

Figure 3: a specific inside view in the source area seems better instead of this extreme general picture over the entire data set. For me is here Figure S5 much better and Figure 3 should remove for Figure S5.

Good point. We initially opted for the combined figure initially, but now realise it obscures some useful information. We have moved the six-panelled figure S5 now to be in the main text, and we have appropriately amended the figure captions. We agree this figure is far easier to interpret.

"The CWTs weighted by Q for each site are plotted in Fig. 4 (CWTs across the whole Arctic region in Fig. S5)"

Figure 4: is very clear and good described in the discussion (starting L253)

Thank you for the kind comment

Figure S1: this is mandatory in the manuscript, and not in the supplementary! But you see the limited data basis and also the lack of data from different station for different time period. This long measurement period from Zeppelin to compare with other station with very limited time span seems critical and should be better discussed in the text. Only 2015 show an observation overlap from 5 station. A specific discussion of this time period is here recommended. Is the result from this period similar / same for the entire period?

Very good point! We include Figure S1 in the text now as Figure 1. We also reformatted the figure so it can fit nicely in one column. We also provide some discussion of both the limited data coverage and the implications for the results, as well as showing the size distributions for these periods of overlap in the following segments, in the methodology:

"There is limited data overlap between the sites, with best overlap during 2015, where data is measured for several months at all sites except one. The mean size distribution from each site for this period, alongside the mean across all time periods is plotted in Fig S1."

In the results:

"Figure S1 shows the average size distribution during the period March – July 2015 where data was being collected at all sites except ALE, where the data for March – July 2013 is shown. All sites have two distinct modes, an Aitken mode peaking somewhere between 20 to 50 nm, and an accumulation mode peaking somewhere from 100 to 200 nm. The average across all periods is also shown, for which the distributions are similar, except ZEP which compared to the whole period of data availability, has a substantially larger mode at ~20 nm in this 2015 period. The size distribution at ALE and VRS shows overall low particle counts, especially at ALE. The two Svalbard sites, GRU and ZEP have similar size distributions, while those at TIK show a large Aitken mode, and UTQ shows a large accumulation mode."

In the discussion:

"These results cover a multi-year period across the Arctic. We highlight that some of these sites have limited data coverage (Figure 1) and the periods of data overlap between sites are limited, although the size distributions for these periods of overlap are similar to the average across all periods (Figure S1). We also note the inherent uncertainty in particle size distribution measurements between sites, especially in both the <20 nm size range, which is particularly important to these NPF studies (Wiedensohler et al., 2012)."

[Figure]

Figure S1: Average size distributions for (A) the months in 2015 with data overlap. ALE shows the data for the equivalent months in 2013, (B) shows the average size distribution for each site for all data.

Figure S3: why as example only type 1,2,3 for Tiksi – is that a typical signature or a special in compare to the stations?

We include the data from TIK here as they are rather typical of Arctic NPF across all the sites. The growth rates are slightly higher than other sites, and the difference between the three types of days is most evident. We include just one site as all of the datasets have bins at different diameters, and this saves us from any uncertainties induced when manipulating the datasets into one diameter scale to plot them on one contour.

L66: It is not enough to write, NPF is a deeply complex process – a little bit more on the theory and main processes would be very helpful

True. We include the following sentences and remove the phrase "complex processes" as it is not a particularly useful wording.

"Different measurements at Arctic sites show a strong annual cycle in aerosol characteristics, largely dictated by new particle formation (NPF) (Tunved et al., 2013; Dall'Osto et al., 2017a; 2018a; 2018b), **a process characterised by a sudden burst of nanometre sizes particles in the atmosphere, followed by their growth to larger sizes. The initial formation of these particles is driven by the clustering of gases in the atmosphere to form clusters at a rate faster than their losses due to evaporation or condensation, the second step is driven by both coagulation, and condensation of vapours with sufficiently low vapour pressures to condense down on new particles (Lee et al., 2019)**"

L81: It is here missing to mention, what are the key parameter for NPF in the Arctic. Are the same like in other regions or not.

We highlight the key parameters in the below section:

"**The key parameters driving NPF in the Arctic are not well understood. In polluted locations the surface area of pre-existing particles often dictates NPF occurrence (Lee et al., 2019), however, in remote locations condensation sinks are consistently low (Sellegri et al., 2019), and concentrations of precursors and solar radiation intensity may be key in dictating NPF frequency and intensity, however**,  with multiple potential mechanisms and many poorly understood sources of precursors from the many and varied eco-regions, **Arctic NPF demands further study**"

L82: The sentence should end with dot, that is missing

This has been fixed

L89-105: The table S1 on the station is not complete, the used instruments is here also recommended, including the specific information, whether the different systems at the different stations means special constrains for the data analysis. How big are the differences between TSI 3034, TSI 3772 CPC and twin DMPS, custom built SMPS and TSI 3010 CPC. This could show very easy in a table…

This is an important thing to highlight. We have no reliable estimates of the uncertainties for each individual instrument, but these will depend on both differences between instruments, and the corrections performed on the data (CPC counting efficiency, DMA transfer function, pipe losses etc.) which are especially pertinent for the smallest size fractions. Previous intercomparison works show that different inversion routines account for some few percent difference in the size distribution, while differences between instruments from different manufacturers are within 10% difference for the 20 – 200 nm size range (Wiedensohler et al., 2012). We include the names of each instrument, as well as the appropriate size ranges measured in Table 1 as well as in the methodology section. Further, we move Table S1 into the main text. We also include a brief discussion of uncertainties in the following sentences

"Intercomparison workshops have shown differences between instruments measuring particle size distributions to be within 10%, increasing at smaller diameters (Wiedensohler et al., 2012). This produces some uncertainty when we are comparing particle formation rates and growth rates of particles in these smaller size regimes, but this uncertainty is substantially smaller than the differences in particle concentrations between sites."

Wiedensohler, A., Birmili, W., Nowak, A., Sonntag, A., Weinhold, K., Merkel, M., Wehner, B., Tuch, T., Pfeifer, S., Fiebig, M., Fjäraa, A. M., Asmi, E., Sellegri, K., Depuy, R., Venzac, H., Villani, P., Laj, P., Aalto, P., Ogren, J. A., Swietlicki, E., Williams, P., Roldin, P., Quincey, P., Hüglin, C., Fierz-Schmidhauser, R., Gysel, M., Weingartner, E., Riccobono, F., Santos, S., Grüning, C., Faloon, K., Beddows, D., Harrison, R., Monahan, C., Jennings, S. G., O'Dowd, C. D., Marinoni, A., Horn, H.-G., Keck, L., Jiang, J., Scheckman, J., McMurry, P. H., Deng, Z., Zhao, C. S., Moerman, M., Henzing, B., de Leeuw, G., Löschau, G., and Bastian, S.: Mobility particle size spectrometers: harmonization of technical standards and data structure to facilitate high quality long-term observations of atmospheric particle number size distributions, Atmos. Meas. Tech., 5, 657–685, https://doi.org/10.5194/amt-5-657-2012, 2012

L114: Is the condensation sinks CS the main parameter identity NPF or the particle growth GR. I think the formulas are for this manuscript secondary. The list of priority for the Arctic site seems more attractive.

As above, we opt to move this table into the main text as we agree that it is important.

L143/144: The explanation, why only type A and type B are used for the identification of NPF, is missing.

We include the following sentence in our methods section

"Formation of particles at the smallest measured sizes is a key characteristic of NPF and is required to calculate formation rates reliably. There is also a chance that Type C events include particles not formed secondarily, but just shows growth of primary particles, and thus we neglect to include Type C events in these analyses"

L184: the discussion of the spatial variability is in general attractive, but the information in the following lines up to 245 is very unstructured. Here a table with the different mean parameter for the three regions makes the discussion on the differences and explanation of reason much easier.

We opt to not include the extra table here as these data are already presented in both figure and the main text, but we expand the discussion of spatial variability in the following

'Here, ALE and VRS are discussed together as "high Arctic", **as both of these sites are high latitude sites with similarly low $J_{10}$, GR, and CS values (Fig. 3),** GRU and ZEP are talked about together as "Svalbard" sites **as they are co-located and surrounded by the same open and ice containing ocean, with similar $J_{10}$, GR, and CS**, and **although dissimilar in $J_{10}$, GR, and CS, the low latitude** TIK and UTQ are seen to represent the "continental Arctic"'

L249/250: I see also a variation of CS at the different sites and a focus to the high frequency period of the NPF could be bring a better inside view.

As mentioned above, the analysis here does indeed focus on the summertime period. We update the sentence in question to include a reference to the condensation sink

"The results reported in this paper highlight the seasonal variation in Arctic NPF (Fig. 2), as well as the variation between different measurement sites during the summertime with $J_{10}$, GR, **CS,** and Q…"

Figure 5: I don't see a big motivation for this figure, could be remove

+

L253-255: what is the motivation for Figure 5 and this sentence? Please explain it.

We include these sentences as well as Figure 5 (now Figure 6) as we feel they accurately represent the complexity of Arctic NPF. We update these sentences as follows

"We show that the vapours which drive particle growth each of these sites often, but not always  **coincide with air masses flowing over particular, directional source regions** (Fig. 4). NPF in the Arctic atmospheric boundary layer is occurring within air masses flowing over vastly different Arctic eco-regions, these being regions of open ocean water, consolidated and open pack ice, snow-covered land, and non-snow-covered land (Fig. 5), **reflected in the variety of mechanisms to be seen in molecular scale measurements of new particle formation and growth (Baccarini et al., 2020; Beck et al., 2020).**"

We also update the figure caption to properly represent what is included in the figure:

"Schematic illustrations of the sea-ice, microbiota, sea-to-air emissions and New Particle Formation (NPF) occurring in the typical summertime stratus-topped Arctic boundary layer. Vertical red and grey bars broadly indicate Cloud Mixed Layer (CML) and Surface Mixed Layer (SML) as inspired by Brooks et al. (2017). The grey box indicates known possible NPF gas precursor from potential Arctic natural terrestrial and marine sources drown below."

Brooks, I. M., Tjernström, M., Persson, P. O. G., Shupe, M. D., Atkinson, R. A., Brooks, B. J. (2017). The turbulent structure of the Arctic summer boundary layer during The Arctic Summer Cloud-Ocean Study. Journal of Geophysical Research: Atmospheres, 122, 9685–9704, https://doi.org/10.1002/2017JD027234

L303-311: Here the message is not clear enough. Figure 1 for example show a high seasonality and for the different parameter not a complete peak for summer, sometime also NPF events in winter. Too much statistics is here not perfect. A specific final statement, what is the key parameter for the NPF event in the Arctic and what are the difference between the three locations would be helpful.

We cannot identify the sole driving factor from this dataset, as it does not seem to be something evident from particle size distribution data alone (such as the condensation sink). The summertime peak in frequency and formation could be due to more intense solar radiation, a lack of Arctic haze, or higher precursor emissions due to melting sea ice and higher biological production in these warmer months, or some combination of all of these. We update our closing statement appropriately

"**While back trajectory analyses can point towards these source regions over long-terms, we still do not know the driving force behind NPF at these sites, as it is likely a combination of precursor emissions, photochemistry, ion pair production, temperature, and pre-existing**

**surface area of aerosol**. Measurements of particle size distributions down to critical cluster size and detailed chemical measurements are required to properly understand NPF at these sites."

---

## Author Response (AR2)

**Collective geographical eco-regions and precursor sources driving Arctic new particle formation, J. Brean et al. Responses to RC1.**

Note: Review comments are displayed in blue, and sections that have been added to the text are coloured *green*. We thank the reviewer for their insightful comments and provide responses below.

**1) For improved readability, the sentence at lines 73-78 could be divided into 3 sentences at the two occurrences of the words 'however'.**

Thank you for the suggestion, this has been done

**2) Figure 1 – please correct BAR to UTQ on the horizontal axis for consistency with the remainder of the manuscript. Also consider adding to the caption a few words to define the abbreviations used to labels the horizontal axis.**

This has been done. We provide the updated figure and caption below

[Figure]

Figure 1: Data coverage for each of the sites. Each individual cell corresponds to one full month of measurements. Fill colour corresponds to the total number of available hourly data as a percentage of the

total hours within that month. **The abbreviations along the bottom axis correspond to Utqiagvik, Tiksi, Mt. Zeppelin, Gruvebadet, Villum research station, and Alert.**

**3) Figure 6 caption, do you mean 'drawn' as opposed to 'drown'?**

Yes. We edit this caption slightly to read better. It now reads as follows

"Figure 6: Schematic illustrations of the sea-ice, microbiota, sea-to-air emissions and New Particle Formation (NPF) occurring in the typical summertime stratus-topped Arctic boundary layer. Vertical red and grey bars broadly indicate Cloud Mixed Layer (CML) and Surface Mixed Layer (SML) as inspired by Brooks et al. (2017). The grey box indicates known possible gas-phase NPF precursors from the potential Arctic natural terrestrial and marine sources drawn below"

**4) Line 43 acknowledges the historical overestimation of low clouds in the models. Please consider if the text should also acknowledge uncertainties in the retrievals of Arctic low cloud coverage e.g., Sect. 4 in Browse, J., Carslaw, K. S., Arnold, S. R., Pringle, K., and Boucher, O.: The scavenging processes controlling the seasonal cycle in Arctic sulphate and black carbon aerosol, Atmos. Chem. Phys., 12, 6775–6798, https://doi.org/10.5194/acp-12-6775-2012, 2012.**

Thank you for the suggestion, this section now reads as follows

"[R]ecently there has been shown a different temporal trend in predicted and observed cloud cover at a high Arctic site (Gryning et al. 2021) continuing a historical overestimation of low-level clouds in the Arctic by climate models, particularly in the wintertime, most particularly in daily data (Quan et al., 2012). Uncertainties also exist in cloud coverage retrieved by satellites, arising from similarities between clouds and ice-snow surfaces, and frequent temperature and humidity inversions (Browse et al., 2012)"

**5) Line 300 appears to be missing the word 'at' before the words 'each of these sites'.**

This has been amended.

**6) Line 343: 'alongside permafrost and precipitation' – is there a word or two missing here? Do you mean 'alongside permafrost thawing and precipitation changes'.**

Yes, thank you. This has been amended.

**7) Line 359: This sentence reads somewhat awkwardly – consider adding the word 'along' before the words 'with sources of organic and inorganic iodine'.**

This has been amended also.